# Learning to count visual objects by combining "what" and "where" in recurrent memory

**Jessica A.F. Thompson**                                    jessica.thompson@psy.ox.ac.uk
**Hannah Sheahan**                                           sheahan.hannah@gmail.com
**Christopher Summerfield**                     christopher.summerfield@psy.ox.ac.uk
*Human Information Processing Lab, Department of Experimental Psychology, University of Oxford*

## Abstract

Counting the number of objects in a visual scene is easy for humans but challenging for modern deep neural networks. Here we explore what makes this problem hard and study the neural computations that allow transfer of counting ability to new objects and contexts. Previous work has implicated posterior parietal cortex (PPC) in numerosity perception and in visual scene understanding more broadly. It has been proposed that action-related saccadic signals computed in PPC provide object-invariant information about the number and arrangement of scene elements, and may contribute to relational reasoning in visual displays. Here, we built a glimpsing recurrent neural network that combines gaze contents ("what") and gaze location ("where") to count the number of items in a visual array. The network successfully learns to count and generalizes to several out-of-distribution test sets, including images with novel items. Through ablations and comparison to control models, we establish the contribution of brain-inspired computational principles to this generalization ability. This work provides a proof-of-principle demonstration that a neural network that combines "what" and "where" can learn a generalizable concept of numerosity and points to a promising approach for other visual reasoning tasks.

**Keywords:** numerosity, counting, gaze, visual reasoning, attention, RNN, dorsal stream

## 1. Introduction

Despite significant recent advances in training deep neural networks to map between text and images, current machine learning models show significant failures in aspects of scene perception that rely on visual reasoning, such as counting how many objects are present in a natural scene (Zhang and Wu, 2020). Even when they are equipped with inductive biases for image segmentation or salience detection, current deep supervised networks often fail to correctly enumerate objects (Lempitsky and Zisserman, 2010; Zhang et al., 2017). Relatedly, state-of-the-art generative models such as DALLE-2 (OpenAI, 2022) struggle to craft images in response to prompts that mandate a specific numbers of objects (Fig. 1). By contrast, most children learn to count objects in the first two years of life, and it remains one of the most robust and ubiquitous human abilities (Schleifer and Landerl, 2011).

In the current paper, we make two contributions. Firstly, we use a stylised dataset to systematically define the conditions under which deep convolutional neural networks succeed and fail at generalizing their ability to count, addressing an issue that remains controversial (Wu et al., 2019). Secondly, we describe a new model whose architecture is inspired by the parallel pathways of the primate visual system, in which visual information flows both ventrally (the "what" stream, to temporal lobe areas that represent objects

in a position-invariant fashion) and dorsally (the "where" or "how" stream, to parietal lobe areas that code for the spatial position, salience and motor affordances induced by objects in a scene) (Goodale and Milner, 1992; de Haan et al., 2018; Bisley and Goldberg, 2010). Like the primate brain, our model apprehends the scene through a series of discrete glimpses, and counting is achieved by combining both the gaze contents ("what") and gaze location ("where") in recurrent memory. We use an approach first proposed by Larochelle and Hinton (2010) in which saccadic output (the movement "afforded" by the objects in the scene) is provided as an additional input to recurrent memory. The network is able to successfully count the number of items in an array and to generalise this ability to a hold-out set containing wholly novel objects. Using a symbolic model, we taxonomize counting problems according to their need for integration of "what" and "where", and show that the advantage of our dual-stream model is greatest when more integration is needed.

Our work builds on the hypothesis that dorsal and ventral streams evolved to factorize representations of the content and structure of sensory experience: the ventral stream encodes object identity while the dorsal stream encodes the abstract structure of space, events, and tasks (Summerfield et al., 2020; O'Reilly et al., 2021; Bottini and Doeller, 2020). We propose that this factorization should enable systematic generalization (e.g., the ability to process new contents via learned structures). Convergent evidence from machine learning similarly points to the value of partitioned representations of content and structure for relational tasks (Kerg et al., 2022; van Steenkiste et al., 2019). Our model thus integrates both ventral and dorsal stream information and we probed its ability to generalize a learned structure (numerosity) to new contents (objects and contexts).

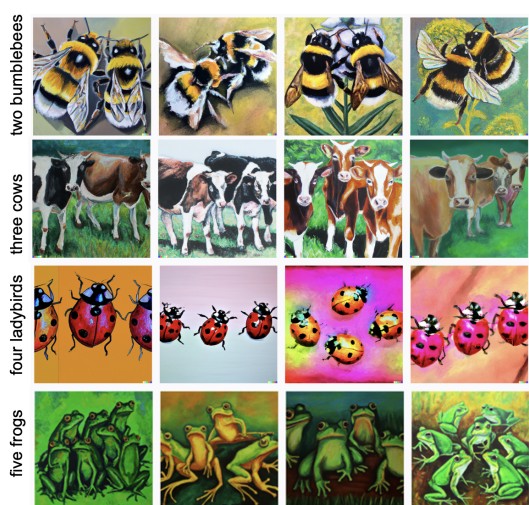

**Figure 1:** Images generated by DALL-E 2 in response to the prompt "A painting of . . ." followed by the text to the left if each row.

## 2. Related Work and Background

**Visual counting in ANNs**   Counting and the neural code for number have been studied in neural networks using three broad approaches. Firstly, some authors have trained standard feedforward convolutional neural networks (CNNs) on annotated datasets to directly report the number of objects in a whole image, equivalent to the human ability to "subitize" or instantly grasp the number of items in an array. This works reasonably well for very small numbers but typically falters as the number of items grows (Segui et al., 2015; Zhang et al., 2017; Chattopadhyay et al., 2017; Fang et al., 2018). Secondly, others have built models that count via explicit object detection and segmentation or density estimation (Lempitsky and Zisserman, 2010; Trott et al., 2018; Zhang et al., 2018). These reports often use attention-based mechanisms to sequentially segment objects using a bounding box.

One challenge is to disambiguate a lone object from two in close proximity (Zhang et al., 2018). Finally, drawing on techniques and assumptions from neuroscience, some authors have sought to identify number-selective units in deep networks that were not trained on number-related objectives (Stoianov and Zorzi, 2012; Nasr et al., 2019). However, these results have often been hard to interpret, especially given that untrained networks seemingly exhibit number neurons (Zhang and Wu, 2020; Kim et al., 2021), casting doubt on the relevance of number selectivity to visual numerosity behaviour. This, in this body of work, there is limited evidence for a general concept of numerosity.

**ANN models of the dorsal stream**   In the wake of the well-documented correspondence between patterns of neural activity in CNNs and the primate ventral stream (Lindsay and Serre, 2021), several researchers have begun looking for a similar correspondence to the dorsal stream. Bakhtiari et al. (2021) and Mineault et al. (2021) both investigate the objective functions that yield dorsal stream-like neural activity. This line of research is related to ours by studying the purpose of the parallel pathways of the primate visual system. However, our goal is primarily to explain a cognitive ability rather than to account for neural activity. Adeli et al. (2022) tackle visual reasoning with a dual-stream, recurrent architecture which takes sequential glimpses based on an object attention mechanism. This model does not process the gaze locations, as in our work. Their model generalizes better to out-of-distribution (OOD) test examples than ResNet18, but still performs much worse on these examples than on in-distribution test images. Fabi et al. (2022) explore another dual-stream recurrent architecture for the compositional problem of learning to write characters.

**Glimpsing computer vision models**   Larochelle and Hinton (2010) describe an architecture that takes a sequence of foveated glimpses based on a learned saccadic policy. This model learns to combine 'what' (gaze contents) and 'where' (gaze location) to solve several object recognition tasks. The Recurrent Attention model similarly processes a sequence of glimpses to obtain an efficient solution to several image classification tasks (Mnih et al., 2014). Inspired by human-like iterative and attentive counting processes, Ren and Zemel (2017) propose a recurrent architecture with visual attention for object instance segmentation. They jointly solve the counting and instance segmentation problems, segmenting one instance at a time.

**Visual numerosity in the primate brain**   In the primate brain, mounting evidence implicates the posterior parietal cortex (PPC) in visual numerosity and in number sense more generally (Nieder and Dehaene, 2009). Monkey electrophysiology has found accumulator neurons as well as neurons tuned to specific numerosities in intraparietal regions (Nieder and Miller, 2004; Roitman et al., 2007). Human neuroimaging tells a similar story, reporting regions tuned to visual numerosity (Piazza et al., 2004) that are arranged topographically in PPC (Harvey et al., 2013, 2015; Harvey and Dumoulin, 2017; Cai et al., 2021). These neural correlates are also consistent with lesion studies in which patients with damage to parietal cortex show deficits in numerical cognition, including visual tasks like dot counting (Ashkenazi et al., 2008; Takayama et al., 1994).

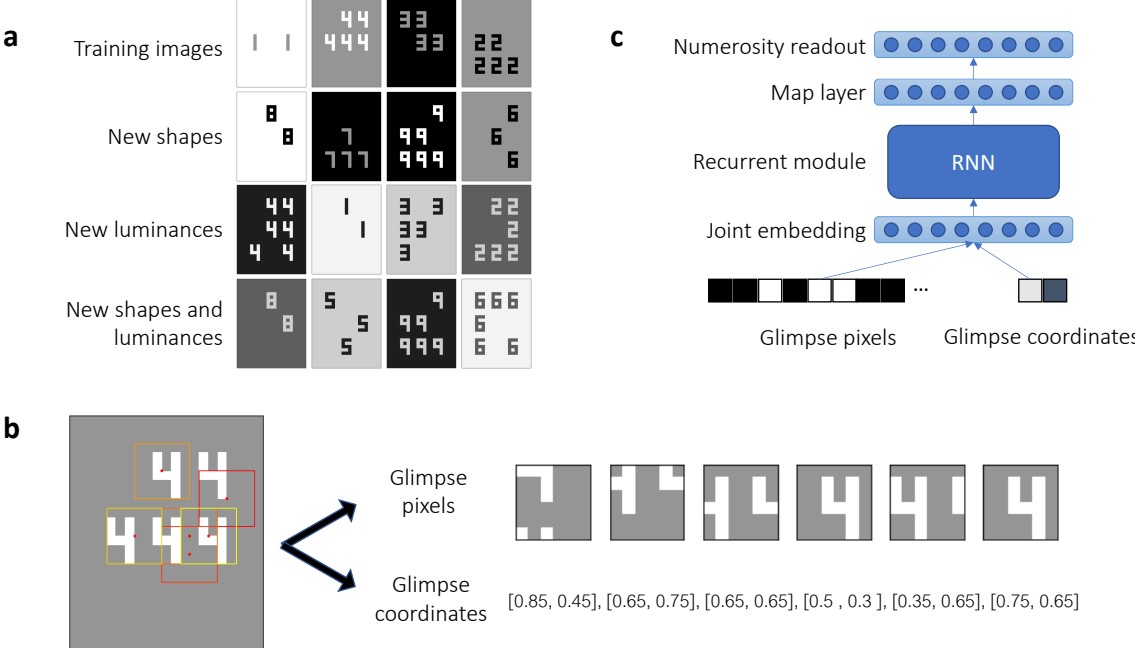

**Figure 2:** Task and model description. (a) All images contain 2-6 items. Models are trained on a subset of item shapes and luminances and tested on held out image parameters. (b) Each image is glimpsed, producing two sequences of observations: the glimpse pixels (gaze contents) and the glimpse coordinates (gaze location). (c) The Recurrent Glimpse Net processes the two glimpse streams to produce a spatial map of the items in the image from which a numerosity label is computed.

## 3. A Neuro-Inspired Recurrent Glimpse Network

### 3.1. The Counting Task

We synthesize grayscale images (27 x 21 pixels) each containing 2-6 items (alphanumeric characters). Items are of constant size and occupy a maximum area of 5 x 3 pixels, and are placed on a uniform background of contrasting luminance. We divide each image into a 3 x 3 grid in which each grid square is a possible item location. We pose the counting task as a supervised classification problem, minimizing a cross-entropy loss between the actual and predicted number of items. During training, foreground and background luminances are sampled randomly from the set {0, 0.5, 1} and item shapes are randomly chosen to be one of the digits 0–4. Model performance is evaluated on a validation set consisting of new images from within the training distribution and three OOD test sets consisting of new shapes (the digits 5–9), new luminances (intermediate grays {0.1, 0.3, 0.7, 0.9}) and both new shapes and new luminances. In our main experiments, all items within an image are the same (homogenous sets) but see Appendix for comparable results with heterogenous sets.

### 3.2. Modeling saccadic trajectories

We call our model the Recurrent Glimpse Net (RGN). Rather than process the entire image in parallel, the RGN receives a sequence of partial glimpses (6 x 6 pixels each) from the image. We used a fixed policy for sampling glimpse locations that obeys the following constraints: (i) there are always six glimpses (the maximum possible number of items); (ii) each glimpse location is drawn from a truncated Gaussian distribution peaking centrally within a grid square containing an item (s.d. $\approx$ 2 pixels, truncation at $\approx$ 4 pixels); and (iii) all items are glimpsed at least once. Whilst this policy is stylised, it approximates that which is obtained by sampling from a visual "salience map", and it replicates many aspects of saccadic behaviour. For example, the Gaussian is sufficiently broad that glimpses frequently span multiple items (or the edges of multiple items), as in Fig. 2b.

### 3.3. Dual Stream Processing

Just as the primate visual system processes "what" and "where", the saccadic glimpsing produces two streams of observations. One contains the sequence of extracted pixels (the 6 x 6 contents of each glimpse) and the other contains the sequence of gaze locations (the [x,y] Cartesian coordinates of the centre of each glimpse). In our task, as in natural vision, neither stream permits accurate counting on its own. Since all items within an image are identical, numerosity information in the gaze contents stream is highly ambiguous. From one glimpse to the next, it is not obvious whether an item (or item fragment) is one that has been previously glimpsed (in which case the count should not be incremented) or is new (in which case it should). Similarly, numerosity is ambiguous from the the gaze location stream alone. For two nearby gaze locations, it is unclear whether the eye is directed towards the same item twice or two different items. Our work tests the contention that these two sources of ambiguity can be resolved by combining information from both streams, and that numerosity judgements will require an integration of "what" and "where".

### 3.4. Network Architecture

The input layer of the network consists of 36 units for the flattened glimpse contents plus two additional units for the gaze locations. These feed into a joint embedding layer preceding a recurrent module. The penultimate layer of processing before the numerosity readout is trained to represent a spatial map of the items in the image via an auxiliary loss. This layer consists of nine units corresponding to the nine possible item locations. The map loss term is a binary cross entropy loss whose target is whether the image contains an item in each of the nine 'slots'. The full architecture is depicted in Fig. 2 and detailed network and training parameters can be found in the Appendix. All models and experiments are coded in Python 3.7.5 using PyTorch 1.11.0.

## 4. Experiments

All models were trained with stochastic gradient descent on 100,000 training images for 500 epochs. Each of the four test sets consisted of 5,000 images. All results are averaged over 10 random initializations. Detailed training parameters can be found in the Appendix.

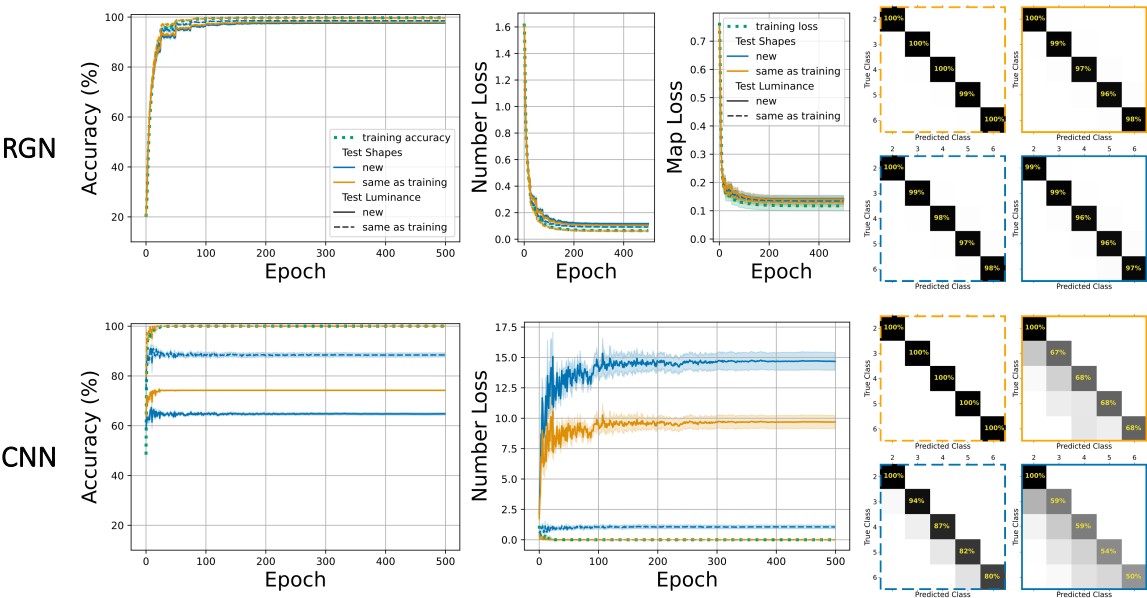

**Figure 3:** (top) Our Recurrent Glimpse Net generalizes to new shapes and new luminances. (bottom) A parameter matched CNN shows significant deficits on all of the OOD test sets. Accuracy and loss curves show mean and SEM over 10 random initializations.

### 4.1. Recurrent Glimpse Net generalizes to new shapes and new luminances

We compare our RGN to a parameter-matched convolutional neural network that receives the entire image as input rather than a sequence of glimpses. The learning curves in Fig. 3 show the performance on the training set and the four test sets. We find that our RGN (top row) is able to generalize to new shapes and new luminances, achieving > 96% accuracy on all test sets. The CNN (bottom row) is able to learn the task but suffers considerably on all the OOD test sets [1].

The remaining analyses are designed to probe how the RGN is able to generalize. Here we are assigning credit to the various computational ingredients and verifying whether the neuroscience- and cognitively-inspired aspects of the model are bearing their expected fruits.

### 4.2. Recurrent Glimpse Net performance depends on the integration of both input streams

The architecture of the RGN is inspired by the premise that numerosity judgements require the integration of "what" and "where". However, this will be more true for some glimpsed images than others, and depends on the precise configuration of items and sampled gaze locations. For some images, the gaze location stream alone could be sufficient to determine

---

1. We note that doubling the size of the CNN did not qualitatively change its behaviour on the OOD test sets.

the numerosity[2]. We therefore wanted to assess how performance depends on the need for integration.

To this end, we developed a manual algorithm that derives numerosity from a symbolic representation of the glimpses. This symbolic model first tries to determine the numerosity from the gaze locations alone, and if successful, assigns the glimpsed image an integration score of 1, indicating that no integration is required. Otherwise, the counter iteratively queries a symbolic form of the glimpse contents to resolve the ambiguity about where items are located. The more times the counter needs to query the glimpse contents, the higher the integration score. The symbolic counter successfully determines the numerosity of 97.24% of the test images. The remaining 2.76% correspond to a small number of edge cases the symbolic model cannot handle and are omitted from the presented analysis. Full details of the symbolic model can be found in the Appendix.

In Fig. 4, we plot performance of three versions of the RGN on the test set consisting of both new shapes and new luminances, split by integration score. Version 1 (left) receives only the gaze locations as input, version 2 (middle) receives only the glimpse contents as input, and version 3 (right) receives both input streams. Neither of the single stream models manages to master the task during training. As-expected, test performance of the locations-only model scales with integration score, achieving 100% accuracy on only those images with an integration score of 1. Only the model with both input streams performs well on all integration scores by the end of training. It masters the 'easier' images faster, requiring more training for the higher integration score images. This confirms that RGN's generalization performance relies on an integration of both input streams, and supports our theory that enactive inputs (gaze locations) can help solve visual reasoning tasks by providing information about the relational geometry among items.

### 4.3. Generalization to new luminances depends on the glimpse coordinate input

Could it be that recurrence alone is sufficient for visual counting in our task? To address this, we use a matched recurrent architecture that, instead of a sequence of glimpses, receives the whole image as input repeatedly. Thus, it lacks the ambiguity in the pixel stream introduced by the partial glimpses, but is also missing the gaze location input present in RGN. This recurrent control fares better than the CNN on the new shapes test set, but still fails to generalize to new luminances (Fig. 5). This suggests that the gaze location input is crucial for learning a representation of numerosity that is invariant to luminance.

### 4.4. A representation of space aids robust generalization in RGN

In a subsequent analysis, we sought to understand the contribution of the spatial map in the penultimate layer. The symbolic model relies heavily on building an approximate spatial map of item locations, i.e. to "fill in" whether an item is present at each candidate location. Notably, biological brains compute spatial maps in the dorsal stream, although their exact computational role is not clear. We thus asked whether this map representation

---

2. Although not present in the datasets analyzed here, there could also be cases where the pixel stream alone is sufficient to determine the numerosity, for example, if all items in an image were distinct and the number of items was equal to the maximum numerosity in the dataset.

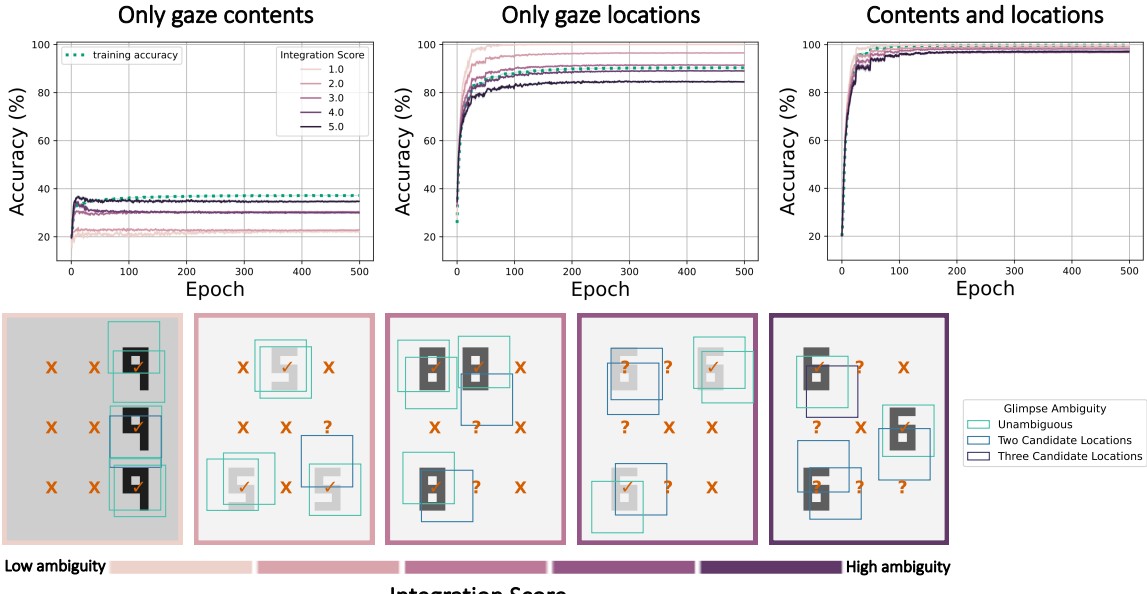

**Figure 4:** (top) Generalization performance on the new shapes and new luminances test set for the RGN which receives only the gaze contents (top left), only the gaze locations (top middle), or both input streams (top right) as a function of integration score. Including both input stream closes the gap between integration scores observed with locations alone. Training performance is included for reference (cyan dashed line). (bottom) Illustration of the symbolic model. Each panel shows an example image with glimpses (blue boxes, coloured to denote ambiguity about item-location assignments) overlaid on items. Panels are arranged from left to right in increasing order of integration score (1-5). The orange ✓, X, and ? symbols indicate spatial locations where the counter respectively determined that there *is*, *is not*, or *could be* an item from the gaze locations alone. For an integration score of 1 (far left), there are no interrogatives because there is no ambiguity about where the items lie.

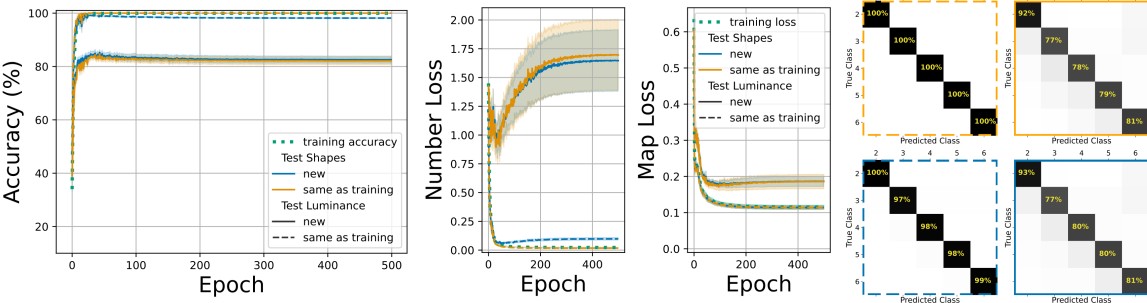

**Figure 5:** Performance of a non-glimpsing recurrent control model of identical architecture as RGN. Without the glimpse coordinate input, this model fails to generalize to new luminances.

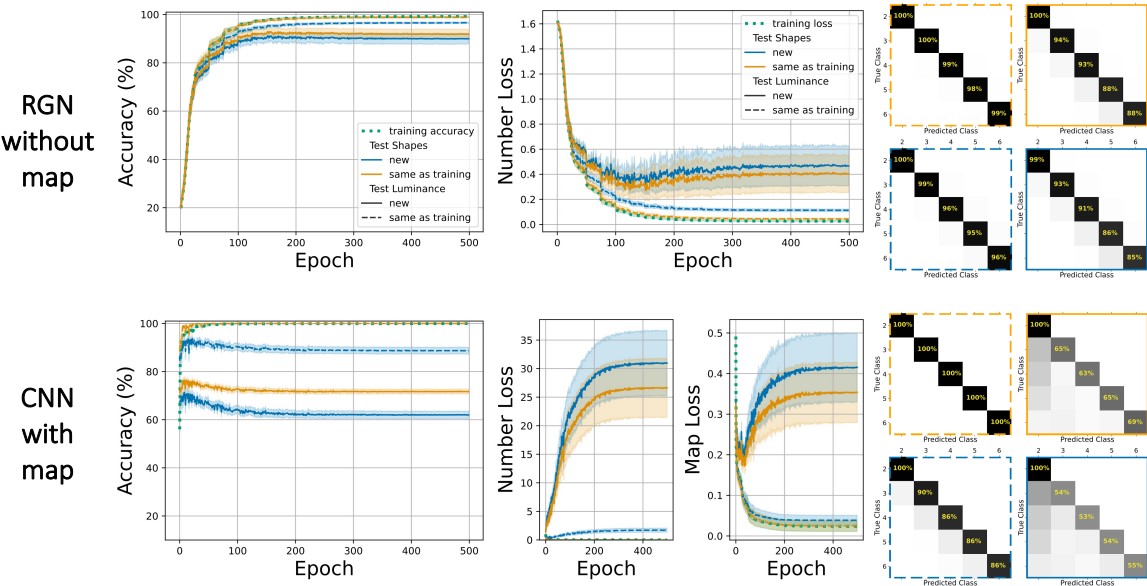

**Figure 6:** (top) Removing the map objective slows learning and impairs generalization in RGN. (bottom) Adding the map loss term to the CNN did not improve generalization.

is necessary and/or sufficient for generalizable numerosity judgements by training a version of RGN without the map objective and a version of the CNN baseline with the auxiliary map objective.

Firstly, we observe that adding the map objective to the CNN baseline was insufficient to render its predictions generalizable. This implies that the map loss is not sufficient for OOD counting. However, removing the map loss term from the RGN objective impairs generalization ability, especially for new luminances. This version of the model learns slower and is more sensitive to random initialization. This implies that the map loss is necessary for accurate counting under the conditions studied here. In the model, the spatial map may help the network organise relational information about the scene (what goes where), which in turn facilitates inferences, such as how many items are present. More generally, these two computational ingredients, the dual stream glimpsing architecture and the spatial map, work together in RGN to learn a generalizable concept of numerosity.

### 4.5. Summary

Fig. 7 summarizes generalization performance on the hardest test set (new shapes and new luminances) for all tested models as a function of the input the model received and whether the map loss term was included in the optimized objective function. The models that generalize best are those that receive the gaze locations as input, but the gaze locations on their own are not sufficient for perfect generalization. When gaze locations are combined with the gaze contents, models can generalize well, but only those models trained with the auxiliary map objective generalize well *consistently*. Without the map objective, generalization performance varies from run to run. Models trained on gaze contents alone perform

only slightly better than chance. Including the auxiliary map objective was not sufficient to enable generalization in models that received the entire image as input.

## 5. Discussion

We designed RGN to embody computational principles hypothesized to contribute to visual scene understanding in the primate brain. We then applied RGN in an idealized and controlled task setting which allowed for careful inspection of generalization behaviour. Our goal was not to train a network to count under naturalistic conditions but to use careful, controlled experiments to study the factors that make counting hard and the computational mechanisms it may require. We compared RGN to several parameter-matched convolutional and recurrent architectures to explore the factors contributing to the RGN's generalization ability, and consequently to provide support for a theory of visual scene understanding.

The critical feature of the RGN is that it combines gaze locations ("where") with gaze contents ("what"). The gaze locations stand in for what could in biology be an overt attention-driven movement, as in a saccade, or a covert attentional signal in the absence

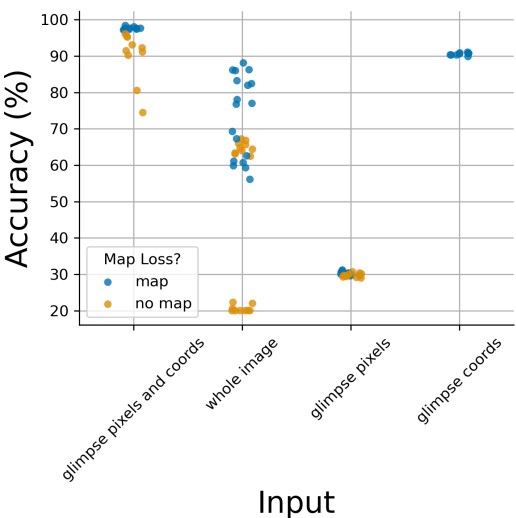

**Figure 7:** Accuracy on the hardest test set as a function of input and whether the auxiliary map objective was optimized. Each point is one model run.

of an explicit movement. In the setting we study, this signal alone is very useful for computing numerosity, but fully accurate performance on new, held-out displays is only possible by integrating information from both streams. We note the caveat that here, we only study a single stylised task under a fixed (if broadly plausible) saccadic policy. More work is required to establish how sensitive our findings are to the specific conditions (e.g., the presence of clutter) and saccadic policy.

Our saccadic policy was designed to yield glimpse sequences that spanned our integration score. We ensured that the vast majority of glimpse sequences received an integration score of 2 or higher, meaning that our symbolic counter required querying both input streams to determine the numerosity. This dataset design allowed us to quantify the specific deficits associated with unistream models. We showed that the performance of a model receiving only gaze locations grades with integration score, generalizing perfectly on glimpse sequences with an integration score of 1 (no integration required). This result validated our interpretation of RGN's generalization success as the result of integrating both input streams. However, it remains to be seen how biologically realistic these saccadic trajectories are. In future work, we plan to explore more biologically plausible models of salience-based attention and to collect eye-tracking while human subjects view images similar to those

used here. Our theory will be further validated if we can predict human visual reasoning responses from real glimpse sequences.

Acquiring the abstract concept of numerosity was one of the Bongard Problems for AI posed over 50 years ago (Bongard, 1970), alongside other relational problems like same-different tasks that still challenge modern AI systems (Stabinger et al., 2021; Mitchell, 2021; Chollet, 2019). In the present work, numerosity recognition is primarily a spatial-relation task. In future work, we plan to extend our approach to other tasks involving an integration of multiple object relations and identities, for example, counting objects that satisfy some query in cluttered images, comparing quantities of different object types, and recognizing Gestalt properties. Such tasks will invariably require a more developed ventral stream, but we expect the computational principles studied here to transfer well to other task settings.

## Acknowledgments

This work was supported by generous funding from the European Research Council (ERC Consolidator award 725937) and Special Grant Agreement No. 945539 (Human Brain Project SGA). Thanks to David McCaffrey and Adam Harris for early discussions.

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

## Appendix A.  Additional Controls

To verify that RGN's generalization ability is not limited to the case of homogenous item sets, we also trained the same model on images with heterogenous item sets. Here the shapes within an image are selected randomly. In comparing Fig. 8 and Fig. 9, we see that the gaze contents are more informative during training when the items are not all the same than when they are, but in a way that does not generalize well to the OOD test sets. The model that receives both inputs still learns to integrate both streams and generalize to new shapes, but struggles somewhat with new luminances compared to when all items are the same, presumably because it has learned to rely more on the gaze contents.

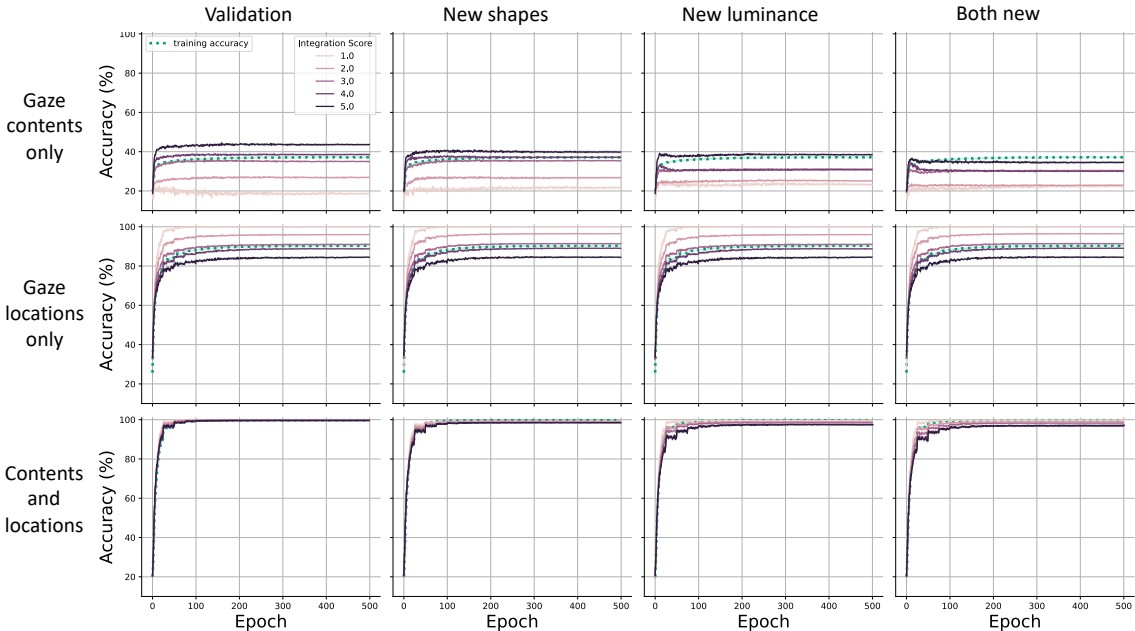

**Figure 8:** Accuracy of uni-stream and dual-stream RGN versions for homogenous item sets divided by integration score.

## Appendix B.  Model and Training Specifications

All models were initialized with random weights and trained for 500 epochs with stochastic gradient descent. We used a momentum factor of 0.9 and a step learning rate schedule, decreasing by a factor of 0.7 every 25 epochs. Dropout (p=50%) was applied before the map layer in all models during training. The CNN included a 2x2 max-pooling layer before the second conv layer. A sigmoid nonlinearity was applied to the map layer units. All other units (except for the number readout units) were leaky rectified linear units with a negative slope of 0.1.

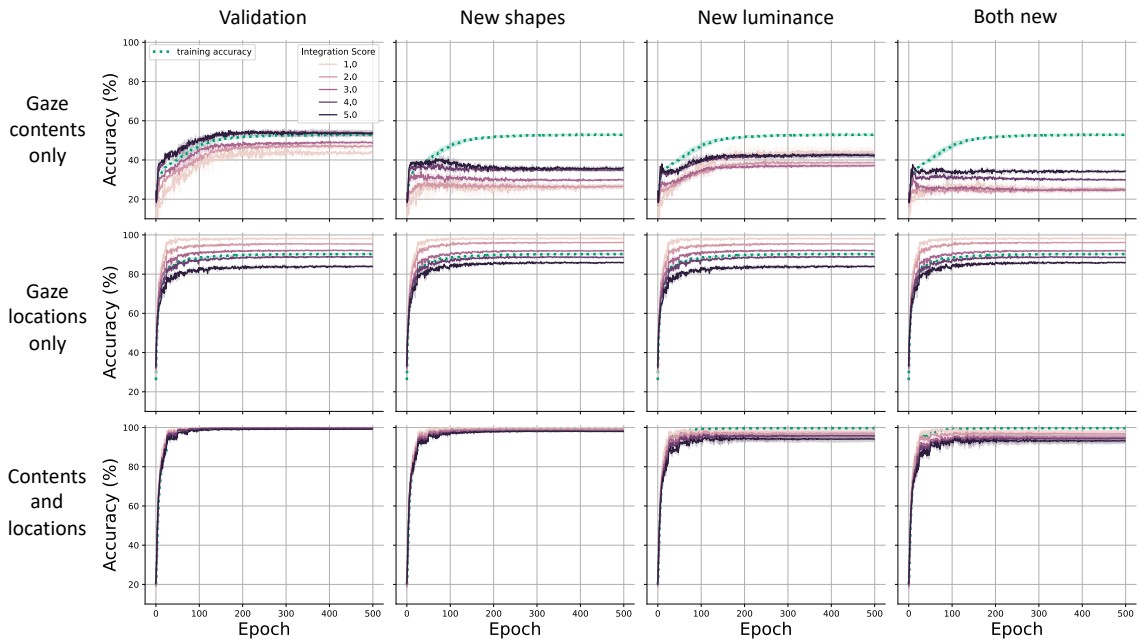

**Figure 9:** Accuracy of uni-stream and dual-stream RGN versions for heterogenous item sets divided by integration score.

**Table 1:** RGN trainable model parameters

| Layer | Shape | Parameters |
|---|---|---|
| embedding | [100, 38] | 3800 |
| i2h | [100, 200] | 20000 |
| h2o | [100, 100] | 10000 |
| map | [9, 100] | 900 |
| number readout | [7, 9] | 63 |
| Total params with biases | | 35079 |

## Appendix C. Symbolic Model of Counting

Here we describe an idealization of the problem of counting the number of objects in an image.

### C.1. Data synthesis

An image consists of a 1x1 square containing $2 \geq N_o \leq 6$ objects. Each object $O_i$ for $i = 1 \ldots N_o$ is described by its Cartesian coordinates $\mathbf{c\_o}_i \in \{(x_i, y_i) | x_i \in S \wedge y_i \in S\}$ where $S = \{0.2, 0.5, 0.8\}$ and its shape. $\mathbf{s\_o}_i$ is a one-hot vector indicating which of $N_s$ shapes $O_i$ is. This defines a 3x3 grid within the image where objects may lie. These nine locations are labeled $0 \ldots 8$. So we can indicate the set of all possible object locations

**Table 2:** CNN trainable model parameters

| Layer | Shape | Parameters |
|---|---|---|
| conv1 | [33, 1, 3, 3] | 287 |
| conv2 | [33, 33, 2, 2] | 4356 |
| conv3 | [33, 33, 2, 2] | 4356 |
| fc1/map | [9, 2904] | 26136 |
| number readout | [7, 9] | 63 |
| Total params with biases | | 35323 |

$A = \{0, 1, 2, 3, 4, 5, 6, 8\}$. For a given image, the set of actual object locations $L$ will be a subset $L \subset A$ where $|L| = N_o$, i.e., the number of 'filled' locations is equal to the image's numerosity.

We then simulate taking a sequence of $N_g$ noisy glimpses of this image. Similarly to the objects, each glimpse $G_j$ is described by its Cartesian coordinates $\mathbf{c\_g}_j$ (gaze locations) and a shape feature vector $\mathbf{s\_g}_j$ (gaze contents) for $j = 0 \ldots N_g$. To generate the gaze locations, we first construct a random sequence of image objects, making sure that the sequence contains every object in the image at least once. For example, in an image that contains 3 objects, our sequence might be $O_2, O_1, O_3, O_3$. Therefore, there is some map $f(j)$ that maps from the glimpse index to the object index. The corresponding sequence of gaze locations will be the coordinates of those objects plus some noise: $\mathbf{c\_g}_j = \mathbf{c\_o}_{f(j)} + \epsilon_j$. The noise for x and y are sampled independently: $\epsilon_{jx} \sim \mathcal{N}(x_{f(j)}, \sigma^2)$ and $\epsilon_{jy} \sim \mathcal{N}(y_{f(j)}, \sigma^2)$. In this version, the noise is truncated such that the total euclidean distance between $\mathbf{c\_g}_j$ and $\mathbf{c\_o}_{f(j)}$ does not exceed some threshold $g(\sigma^2)$, but this could be relaxed in a fully Bayesian version of the counter. The glimpse shape feature $\mathbf{s}_j^{(g)}$ is also constructed from the object shape features. Whereas the $m^{th}$ element of $\mathbf{s\_o}_i$ indicates whether object $i$ is an instance of shape $m$, the $m^{th}$ element of $\mathbf{s\_g_j}$ is the proximity of glimpse $j$ to objects of shape $m$.

$$\mathbf{s\_g}_j^{(m)} = \begin{cases} \sum_{k \in K} 1 - \frac{dist(\mathbf{c\_g}_j, \mathbf{c\_o}_k)}{g(\sigma^2)} & \text{if } |K| > 0 \\ 0 & \text{otherwise} \end{cases}$$

(1)

where $K$ is the set of object indices for which $\mathbf{s\_o}_k^{(m)} = 1$ and $dist(\mathbf{c\_g}_j, \mathbf{c\_o}_k) \leq g(\sigma^2)$. This proximity score is a positive real number where 0 indicates that there are no objects of shape $m$ in the vicinity of that glimpse and 1 could mean either that the gaze locations are equal to the coordinates of an object of shape $m$, or that a glimpse is in between two objects of shape $m$. If the glimpse is close to more than two objects of the same shape, this proximity value can exceed 1.

## C.2. Counter

Given only a sequence of glimpses $G_j = (\mathbf{c\_g}_j, \mathbf{s\_g}_j)$ for $j = 1 \ldots N_g$, the noise level $\sigma^2$ with which the glimpses with synthesized, and knowledge of the minimum and maximum numerosities, the counter tries to determine the number of objects in the image. It will do so by estimating $L$, the set of object locations, or by taking a shortcut based on prior knowledge about the minimum and maximum possible number of objects, if available. The counter will assign each image a difficulty level (or integration score) ranging from 0–6 which indicates the degree to which the two input streams (gaze locations and gaze contents) need to be integrated in order to determine the numerosity.

### C.2.1. Shortcuts

There are two shortcuts that might be available, one based on $s\_g$ and one based on $c\_g$. The number of unique shapes represented in $s\_g$ provides a lower bound on the number of objects in the image. Our lower bound is initialized to the minimum numerosity. If the number of unique shapes is greater than this initial lower bound, we will update the lower bound to be equal to the number of unique shapes. When the number of unique shapes is equal to the maximum numerosity present in the dataset, the numerosity can be predicted exactly without further processing. Images for which this shortcut is available are assigned a difficulty level of 0. If this shortcut is unavailable, the next step is to process $c\_g$.

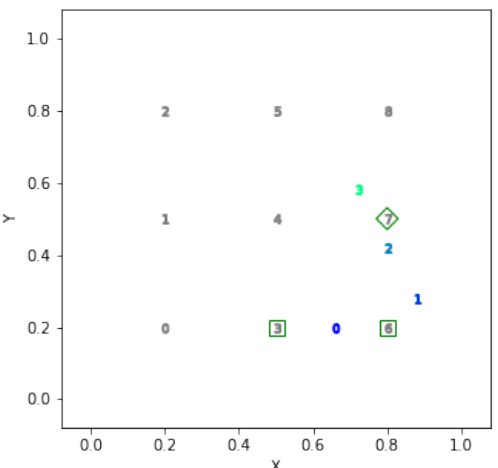

**Figure 10:** An idealized image depicting the nine possible object locations. In this example, the image contains three objects: a square at location 3, another square at location 6, and a diamond at location 7. Glimpse locations are indicated by the blue-green numerals.

### C.2.2. Processing gaze locations

For each glimpse coordinate tuple $c\_g$, we infer a set of candidate object locations. This will be the set of object locations within some radius (defined by the noise level) of the glimpse coordinate. For example, in Figure 10, the set of candidate locations for $G_0$ would be $3, 6$; $G_0$ indicates that there could be an object in location 3, in location 6, or in both location 3 and 6. With these sets of candidates for each glimpse, we can now check for our second shortcut. If the total number of unique candidate locations over all glimpses is equal to the lower bound, then the numerosity is equal to the current lower bound. Images for which this shortcut are available receive an integration score of 1.

Based on the candidate sets, each glimpse is categorized as either unambiguous—if there is only one location in its set of candidates—or ambiguous, otherwise. Without any further processing, the unambiguous glimpses tell us locations where there are definitely objects. We will update our estimate of the object locations $\hat{L}$ accordingly. The ambiguous glimpses

can be further categorized as those that need to be resolved to determine the numerosity and those that do not. The ambiguity of an ambiguous glimpse does not need to be resolved if its set of candidate locations is contained within our current estimate $\hat{L}$. For example, in Figure 11, glimpses 1 and 3 are unambiguous, as indicated by the filled line that connects them to only one location. Glimpses 0 and 2 are both ambiguous, but only the ambiguity of glimpse 0 needs to be resolved because the candidates of glimpse 2 (indicated by the dashed lines) are locations already indicated by the unambiguous glimpses. If there are no ambiguous glimpses that need to be resolved, then we are done: our current estimate $\hat{L}$ but be equal to $L$ and the numerosity must be $|\hat{L}|$. Images with no ambiguous glimpses to be resolved also receive a difficulty level of 1 because it required a single pass of computation through $c\_g$ to determine the numerosity.

### C.2.3. Resolving ambiguity with the shape feature

If there are ambiguous glimpses to be resolved, we use $s\_g$ to resolve that ambiguity. $s\_g$ is used to arbitrate among the hypotheses proposed by the initial processing of $c\_g$. For example, recall in Figure 11, glimpse 0 is an ambiguous glimpse that needs to be resolved to determine the numerosity. Its candidate locations are 3 and 6. Location 6 has already been surmised to host an object based on information from the other glimpses. So the hypotheses are :

1. There is only an object in location 6. Location 3 is empty.

2. There are objects in both location 3 and location 6.

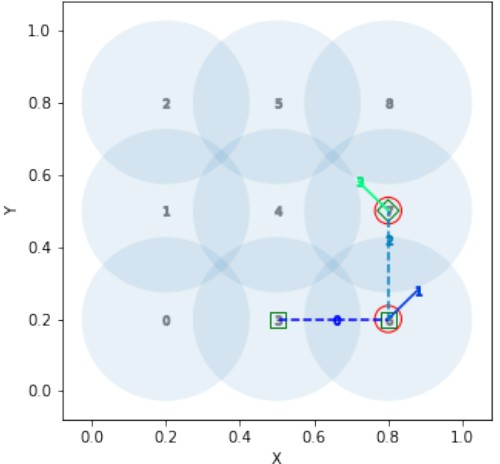

**Figure 11:** Knowledge state after processing the gaze coordinates. Filled lines indicate candidate locations for ambiguous glimpses. Filled lines indicate unambiguous glimpses. Red circles indicate locations where items are known to lie. $G_1$ and $G_3$ unambiguously reveal that there are items in locations 8 and 7 respectively. The candidate locations for $G_2$ are also 7 and 8, so $G_2$ is ambiguous but irrelevant. The only glimpse ambiguity that needs to be resolved is that of $G_0$.

These hypotheses predict different patterns in $s\_g_0$. If hypothesis 1 were true, $s\_g_0$ would be all zeros except for one element which would be equal to the proximity of glimpse 0 to location 6. There are two ways that hypothesis 2 could be true. Either location 3 and 6 host the same type of shape, or they host different shapes. If two different shapes, $s\_g_0$ will be all zeros except for two elements, whose values are equal to the glimpse's proximity to location 3 and 6. If the same shape, than $s\_g_0$ will be all zeros except for one element whose value will be equal to the sum of the proximities of glimpse 0 to location 3 and location 6. These three predictions can be compared to the actual $s\_g_0$ to

determine which hypothesis is true. In general, the counter enumerates all hypotheses suggested by $c\_g_j$ and uses $s\_g_j$ to arbitrate among them. When the set of candidate locations is greater than 2 (can be max 4 in these simulations), the number of predicted patterns increases significantly since we need to enumerate all possible combinations of shapes.

After resolving an ambiguous glimpse, we update $\hat{L}$ and check if any of our previous stopping conditions are met. Note that resolving one ambiguous glimpse might render other ambiguities inconsequential as well, so we re-evaluate the set of ambiguous glimpses to be resolved. If any such glimpses remain, we pick another at random to be resolved and go back to the beginning of the loop. The integration score will be incremented each time we resolve the ambiguity of an ambiguous glimpse. A glimpse cannot ever necessarily be uniquely assigned to an object—a glimpse is not 'of' an object per se—but each glimpse, because of how they were simulated, contains information about the set of objects in the image.

