# OpenReview forum: "Learning to count visual objects by combining "what" and "where" in recurrent memory"
_NeurIPS.cc/2022/Workshop/GMML — Gaze Meets ML 2022 Poster_

### Official Review · Reviewer_PLjg · 2022-10-14
**Review on Learning to count visual objects by combining "what" and "where" in recurrent memory**

**Rating:** 7
**Confidence:** 4

**Review:**

## Paper Summary
The authors tried to improve the accuracy of object counting in images by combining gaze location and gaze contents. The authors state a proof-of-principle paper since the experiments are not done under naturalistic conditions (authentic images) but instead constructed to emphasize the requirements of the difficult task of counting in images. Their model intelligently adds gaze content (glimpse content) based on an integration score if gaze location (glimpse location) is not enough. Thus, the model is efficient and avoids ambiguities.
The results show that their approach achieves high accuracy values and is able to better generalize than other models.

## Quality
The paper is well written, yet complex and not easy to understand. The structure of the paper is unconventional since the section titles read as a summary of the following text. The section titles are sometimes full sentences (especially 4.1, 4.2, 4.3, and 4.4).
Section 4 also contains results and partly discusses results.
I encourage the authors to better structure the work into Methods (containing model description and experimental setups), Results (ONLY containing results), and Discussion. The reason is the general discussion at the end of the paper. Thus, a dedicated discussion section makes sense.

The paper needs spell-checking and proofreading. Examples:
Line 24: ... a specific numbers of objects ...
Caption Figure 1: ... the text to the left if each ...
Line 62: Abbreviation ANN was never introduced
Line 140: ... numerosity is ambiguous from the the ...
Line 142: ... or two different items ,, (two dots at the end of the sentence)
Caption Figure 3: Abbreviation SEM was never introduced
Line 194: To address this, we use (wrong tense --> I suggest "used")

The appendix is almost the length of the paper.

## Clarity
Despite the unconventional structure of the paper, the authors are overall clear about their work.
The authors state that the tests were not done under naturalistic conditions (authentic images) to emphasize the requirements of the difficult task of counting in images. However, these requirements were not well worked out. If the authors insist on non-naturalistic conditions to achieve a proof-of-principle, the resulting requirements/factors should be presented more clearly and highlighted as one of the major contributions.

## Originality
The literature research is well presented and the research gap is clear. The method is aim-directed and is based on neurophysiological principles which in turn were confirmed with literature.
Thus the approach seems novel and has not yet been published.

## Significance
The work is (to me) of high significance for the community since generalization seems to be addressed very well.
The work has to be tested under naturalistic conditions. Thus, one criticism is why the models were not tested under naturalistic conditions since the generalization could have at least led to superior results compared to the baseline. See also comment under *Clarity*

## Final Statement
The paper's method is interesting and of high significance to the community. Had there been fewer grammatical and spelling errors, a better structure, and better highlighting of contributions, the paper would have been at least a clear accept.

---

### Official Review · Reviewer_okFR · 2022-10-17
**Too simplistic**

**Rating:** 3
**Confidence:** 3

**Review:**

The authors propose to use a two input recurrent neural network to count the number of objects in an image. For this, they use a sequence of randomly selected bounding boxes as one input and the coordinates of the bounding boxes as another input.They conclude that the use of these two sources of information as "what" (the bounding box image) and "where" (the bounding box coordinates) is superior that either of the inputs alone in the task of counting the objects.

Unfortunately, the paper over-promises and then under-delivers. To run an experiment, they use such an unrealistic and oversimplified data set that does not allow to look at any of the conclusions with confidence. In addition, even though they mention many prior art papers that deal with object counting, they fail to compare their method to any of them or show that their method does not have any of the shortcomings of the previous methods.

I recommend that the authors train and test their model on a more realistic data set with various objects on varying background other than 5 numbers on a grid over a uniform background.

Also, I have one question: If figures 3-6 show test results, why do you have epochs there? The test results should not vary based on epoch.

---

### Official Review · Reviewer_x5n4 · 2022-10-19
**Interesting study on object counting**

**Rating:** 7
**Confidence:** 2

**Review:**

The paper taclkes the task of automatically counting objects in a scene.
The work studies when convolutional neural networks succeed and fail at generalizing
their ability to count objects in a scene.
The paper is well written and provides good material to anyone that wants to approach the study of the task.
It is a reasonable workshop submission.

---

### Meta-Review · Area_Chair_WNeR · 2022-10-20

**Recommendation:** Accept (Poster)
**Confidence:** 5

**Metareview:**

This paper describes a careful ablation study to investigate how neural networks inspired by primate visual system on two pathways, ventral and dorsal, can generalize in the difficult task of numerosity. This is an interesting line of research as it can pave the way for deeper understanding of human visual reasoning. While the experimentations could be designed more carefully (i.e. one reviewer's point on dataset simplicity) the contributions of this work should be neglected and should be shared with the broad community.

---

### Decision · Program_Chairs · 2022-10-20

Accept (Poster)